# Cell fate potentials and switching kinetics uncovered in a classic bistable genetic switch

Xiaona Fang[1,2,3,4], Qiong Liu[1], Christopher Bohrer[2], Zach Hensel[2,5], Wei Han[3], Jin Wang[1,3,4] & Jie Xiao[2]

Bistable switches are common gene regulatory motifs directing two mutually exclusive cell fates. Theoretical studies suggest that bistable switches are sufficient to encode more than two cell fates without rewiring the circuitry due to the non-equilibrium, heterogeneous cellular environment. However, such a scenario has not been experimentally observed. Here by developing a new, dual single-molecule gene-expression reporting system, we find that for the two mutually repressing transcription factors CI and Cro in the classic bistable bacteriophage λ switch, there exist two new production states, in which neither CI nor Cro is produced, or both CI and Cro are produced. We construct the corresponding potential landscape and map the transition kinetics among the four production states. These findings uncover cell fate potentials beyond the classical picture of bistable switches, and open a new window to explore the genetic and environmental origins of the cell fate decision-making process in gene regulatory networks.

[1] State Key Laboratory of Electroanalytical Chemistry, Changchun Institute of Applied Chemistry, Changchun 130022, China. [2] Department of Biophysics and Biophysical Chemistry, Johns Hopkins School of Medicine, Baltimore, MD 21205, USA. [3] College of Physics, Jilin University, Changchun 130012, China. [4] Department of Chemistry and Physics, Stony Brook University, Stony Brook, NY 11790, USA. [5] Present address: Instituto de Tecnologia Química e Biológica António Xavier, Universidade Nova de Lisboa, Av. da República, 2780-157 Oeiras, Portugal. Correspondence and requests for materials should be addressed to J.W. (email: jin.wang.1@stonybrook.edu) or to J.X. (email: xiao@jhmi.edu)

Cell fate decision-making is the process of a cell committing to a differentiated state in growth and development. The decision is often carried out by a select set of transcription factors (TFs), the expression and regulatory actions of which establish differentiated programs of gene expression[1]. Bistable switches, which consist of two mutually repressing TFs, are the most common gene regulatory motifs directing two mutually exclusive gene expression states, and consequently distinct cell fates[2–12]. Theoretical studies suggest that the simple circuitry of bistable switches is sufficient to encode more than two cell fates due to the non-equilibrium, heterogeneous cellular environment, allowing a high degree of adaptation and differentiation[13–16]. However, new cell fates arising from a classic bistable switch without rewiring the circuitry have not been experimentally observed[17–19].

Here, by developing a dual single-molecule gene-expression reporting system, we demonstrate experimentally the emergence of two new expression states in the model bistable switch of the bacteriophage λ[20]. We construct the corresponding potential landscape and map the transition kinetics between the four production states, providing insight into possible state-switching rates and paths. These findings uncover cell fate potentials beyond the classical picture of λ switch, and open a new window to explore the genetic and environmental origins of the cell fate decision-making process in gene regulatory networks.

## Results

**Construction and validation of DuTrAC.** The λ switch is composed of two mutually repressive TFs, CI and Cro (Fig. 1a); the expression of CI but not Cro confers lysogenic growth, and the expression of Cro but not CI confers lytic growth. The λ switch has served as a paradigm for studying gene regulation and cell fate determination[9–11,20,21], but the real-time switching kinetics and paths between the two distinct, mutually exclusive gene expression states have not been elucidated experimentally. To achieve these goals, we developed a dual single-molecule gene-expression reporting system to follow the stochastic expression

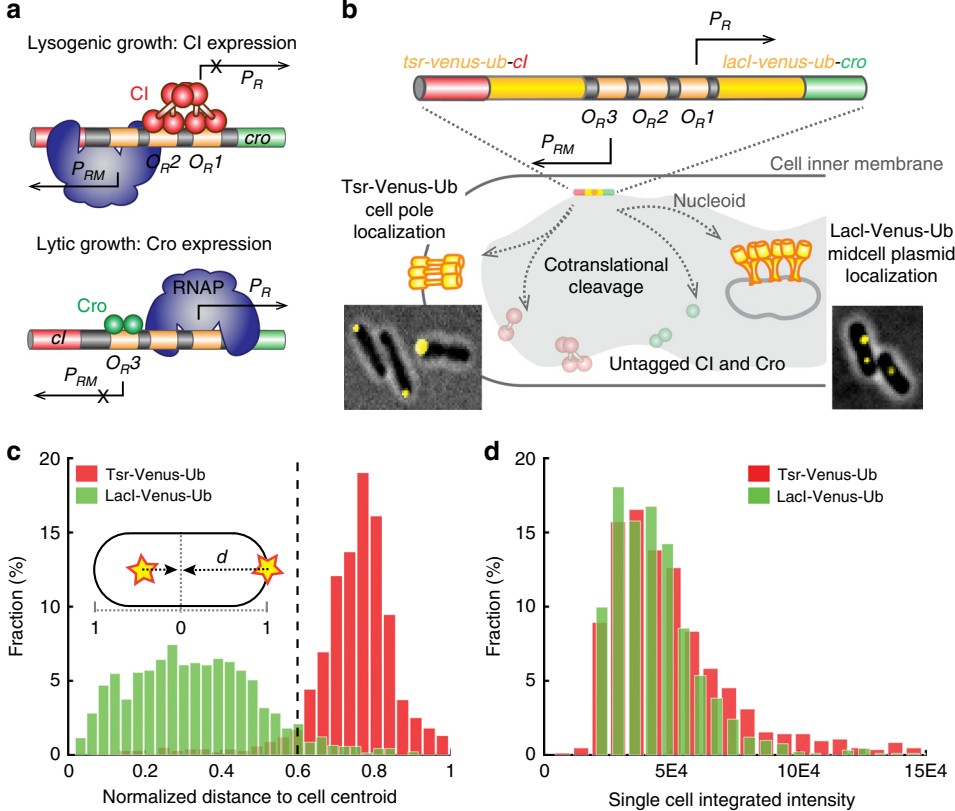

**Fig. 1** Validation of the dual single-molecule gene-expression reporting system DuTrAc. **a** Schematic drawing of the two gene expression states of the λ genetic switch. In lysogenic growth, the λ repressor CI binds to its operator sites $O_R1$ and $O_R2$ to stimulate its expression from promoter $P_{RM}$, and at the same time shuts down Cro expression from promoter $P_R$. In lytic growth, Cro binds to $O_R3$ to repress CI and turns on its own expression. **b** Schematic drawing of the DuTrAC system. The λ switch (colored bar) containing *tsr-venus-ub-cl* and *lacl-venus-ub-cro* is integrated into the chromosome at the *lac* operon locus. The expressed fusion polypeptide chain is cotranslationally cleaved by the constitutively expressed deubiquitinase UBP1 at the last residue of the Ub sequence, separating cell-pole-targeting Tsr-Venus-Ub from CI, and mid/quarter-cell-targeting Lacl-Venus-Ub from Cro for single-molecule detection. CI and Cro are thus untagged and can bind respective operators to regulate gene expression. Representative cell images of pole- or midcell-localized Venus fluorescence spots are shown as insets. **c** Histograms of normalized distance (*d*) of Venus fluorescence spots to cell centroid (inset) in strain XF002 (red, expressing Tsr-Venus-Ub-CI only, Supplementary Figure 1, Table 1) and XF003 (green, expressing Lacl-Venus-Ub-CI only, Supplementary Figure 1, Table 1) in the presence of the deubiquitinase UBP1; 92% of Tsr-Venus-Ub spots (*n* = 993 spots) localized at *d* ≥ 0.6 (dashed line); 95% of Lacl-Venus-Ub spots (*n* = 1384 spots) localized at *d* < 0.6, suggesting that the threshold of *d* = 0.6 could be used to distinguish the identity of the fused protein. **d** Histograms of integrated fluorescence level of Tsr-Venus-Ub (red) and Lacl-Venus-Ub (green) in individual XF002 and XF003 cells. The distributions and mean levels (4.8 ± 2.9 × 10$^4$, *n* = 840 cells for XF002, and 4.2 ± 2.1 × 10$^4$, *n* = 913 cells for XF003, *μ* ± s.d.) of Venus fluorescence in the two strains were indistinguishable from each other, indicating that both Tsr-Venus-Ub and Lacl-Venus-Ub, despite the different mRNA and protein sequences, reported the expression levels of CI equivalently

dynamics of CI and Cro simultaneously in the same cells (Fig. 1b).

In the dual gene-expression reporting system, we fused a fast-maturing yellow fluorescent protein variant, Venus[22], to one of two cellular localization tags, Tsr or LacI, to distinguish the production of CI and Cro in the same cell. The strategy of using two different subcellular localizations differs from previous studies using fluorescent proteins of different colors[23], and avoids the major disadvantage of temporal mismatches caused by different fluorescent protein maturation rates (e.g., ~1 h for red fluorescent proteins such as mCherry[24] and ~5–10 min for Venus[22,25–28]). Tsr is a membrane protein that localizes rapidly and specifically to cell poles[29]. LacI binds specifically to 256 lacO sites (lacO[256]) incorporated onto a multi-copy, mid/quarter-cell-localizing RK2 plasmid pZZ6[30]. With the ability to localize single fluorescent protein molecules with 30–40 nm precision in live Escherichia coli cells[27,31], we could distinguish between individual Tsr-Venus and LacI-Venus molecules based on their subcellular positions. Using a control strain XF004 expressing Tsr-Venus and LacI-mCherry independently (Supplementary Figure 1, Supplementary Table 1), we demonstrated that there was indeed minimal spatial overlap (~2%) between the two localization tags (Supplementary Figure 2 and Supplementary Movie 1).

Next, to distinguish the expression of CI and Cro in the same cell while avoiding possible disruptions of their functions due to the fluorescent protein fusion, we generated two translational fusion genes, tsr-venus-ub-cI and lacI-venus-ub-cro, and used the CoTrAC strategy (CoTranslational Activation by Cleavage[26,27]) to cleave cotranslationally the Tsr-Venus-Ub or LacI-Venus-Ub reporter from CI or Cro (Fig. 1b). This strategy ensures a 1:1 ratio in real-time between localized Venus reporter molecules and the fused CI or Cro molecules. Using two control strains expressing only Tsr-Venus-Ub-CI (XF002) or LacI-Venus-Ub-CI (XF003), we verified that the cellular localization and fluorescence intensity of cell-pole- and midcell-targeted Venus spots faithfully reported both the identity and expression level of CI (Fig. 1c, d, Supplementary Figure 1, Supplementary Movies 2 and 3). We named this new, dual gene-expression reporting system DuTrAC (Dual coTranslational Activation by Cleavage).

**λ switch exhibits two new CI and Cro expression populations.** To investigate the regulatory dynamics of CI and Cro in the λ switch using DuTrAC, we constructed strain XF204. We fused tsr-venus-ub to a temperature-sensitive CI mutant (cI857[32]), and lacI-venus-ub to cro, replacing the native cI and cro genes in the genetic switch, similar to what was previously described (Supplementary Figure 1, Supplementary Table 1)[27]. We then integrated this circuit from $O_L$ to the end of the cro gene into the chromosome of E. coli MG1655 strain at the lac operon locus (Fig. 1b). Hence, Tsr-Venus-Ub reports the expression of CI857, and LacI-Venus-Ub reports the expression of Cro. We used the temperature-sensitive mutant CI857 (A66T[32]) in place of wild-type (WT) CI[WT] for the convenience of using temperature to tune the fraction of active CI. CI857 has normal DNA-binding affinity and transcription regulation activity at the permissive temperature of 30 °C[33]. At higher temperatures, an increasing fraction of CI857 becomes inactivated due to misfolding and subsequent degradation[33], and therefore temperature can be used as a convenient "control knob" to change the fraction of active, WT CI molecules[8]. We verified that the fusion of DuTrAC reporters to CI and Cro did not change the switching behavior of the genetic switch (Supplementary Figure 3A). Furthermore, to examine the switching behavior across different protein expression levels, we generated two additional strains XF214 and XF224 (Supplementary Figure 1, Supplementary Table 1), in which the expression level of LacI-Venus-Ub-Cro was reduced in the order of $[Cro]_{XF224} < [Cro]_{XF214} < [Cro]_{XF204}$. Using Western blotting, we confirmed that the expression levels and switching behaviors of these strains were as expected (Supplementary Figure 3B). Note that in the following experiments, for simplicity, we referred CI857 as CI.

To investigate the switching behavior of the modified λ switch, we first quantified the expression levels of CI and Cro in individual cells of strain XF204, XF214, or XF224 maintained constantly at different temperatures for >20 generations (Fig. 2, Supplementary Figures 4, 5 and Supplementary Table 2). We found that consistent with the typical bistable behavior, at a low temperature (30 °C), cells had few Cro but predominately CI molecules (hereafter termed [L, H] for low-Cro and high-CI level,

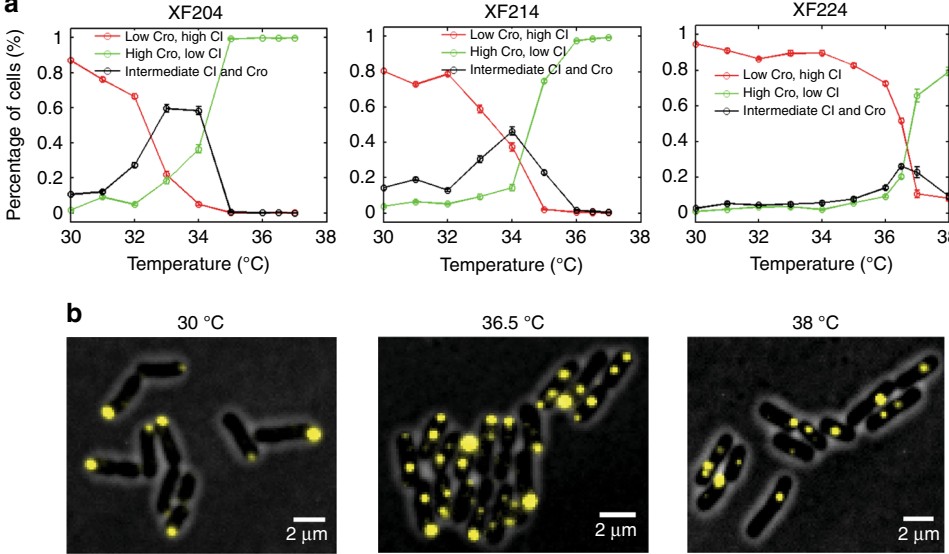

**Fig. 2** Expression levels of CI and Cro in strains XF204, XF214, and XF224 at different temperatures showed more than two expected cell populations. **a** Percentages of cells having CI only (red, copy number ratio $r = CI/(CI + Cro) \geq 0.8$), Cro only (green, $r \leq 0.2$), or both CI and Cro (black, $0.2 < r < 0.8$) in strains XF204, XF214, and XF224 at different temperatures. **b** Representative fluorescent images of XF224 cells showing CI expression (yellow pole-localized Tsr-Venus-Ub spots) and Cro expression (yellow quarter/midcell-localized LacI-Venus-Ub spots) at low, intermediate, and high temperatures overlaid with phase-contrast cell images (gray)

Fig. 2a, red curves); at a high temperature (37 °C) the switch was flipped and cells predominately existed in high-Cro, low-CI level ([H, L], Fig. 2a, green curves). Interestingly, between the two extreme temperatures, we observed that a large population of cells had both CI and Cro [H, H] in the same cells at intermediate levels (Fig. 2a, black curves, Supplementary Figure 5, Supplementary Table 2). Reduced Cro levels in strains XF214 and XF224 did not abolish the presence of [H, H] cells, but shifted the temperature at which the percentage of this population of cells was the highest from 33 °C in XF204 to 34 °C in XF214 and to 36.5 °C in XF224 (Fig. 2, Supplementary Figure 5, Supplementary Table 2). A fourth population of cells having little CI or Cro ([L, L]) also existed at these temperatures. A few representative images of the four cell populations of strain XF224 at low, intermediate, and high temperatures are shown in Fig. 2b. Previous studies probing CI and Cro expression levels independently did not observe the presence of the two new populations of cells[9,34].

**Observing the switching of CI and Cro production**. The observation of cells having both CI and Cro indicated that cells could switch between CI- and Cro-expressing states within each other's degradation time scales, and/or there existed a new expression state in which CI and Cro were expressed concurrently. The snapshot nature of the above measurement could not distinguish these possibilities. In addition, the snapshot measurement of CI was complicated by the fact that at high temperatures an increasing population of CI857 becomes inactive; hence, the actual steady-state level of active CI (molecules/cell) is only proportional to the measured level of Tsr-Venus-Ub. These problems could be circumvented by following protein production in real time; the number of newly produced protein molecules per unit time directly reflects promoter activity during that time without the convolution of any downstream processes. Therefore, we grew XF224 cells in a precision temperature-control chamber ($T = 36.5 \pm 0.1$ °C over the length of the experiment of ~7 h, Supplementary Figure 6) on a microscope stage, and counted the number of newly produced CI and Cro molecules in individual cells every 5 min for multiple generations (Supplementary Figure 7, Supplementary Movies 4, 5, mean cell cycle time $\tau = 71 \pm 22$ min, $\mu \pm$ s.d., $n = 457$ cell cycles). We photobleached Venus molecules after each detection, so that new fluorescent molecules detected after 5 min of the dark interval were newly produced during the 5 min[26,27]. We chose strain XF224 because it had the lowest Cro steady-state levels compared to XF204 and XF214 (Supplementary Figures 3, 5), facilitating the accurate identification and counting of single Venus molecules in small *E. coli* cells (Supplementary Figure 8). We choose to conduct the real-time experiment at 36.5 °C because the steady-state experiment showed that at this temperature XF224 has the largest population of cells expressing both CI and Cro.

In Fig. 3, we show four representative time traces of different XF224 cell lineages. For each colony, we only picked randomly one cell lineage for analysis in order to avoid double-counting data. We observed stochastic, anti-correlated production of CI and Cro (Fig. 3a–d, Supplementary Figure 9). Intriguingly, in many time traces, we observed that there were periods of time in which neither CI or Cro was produced, or both were produced. The presence of the four production populations was evident when we plotted the two-dimensional (2D) histogram of the number of CI and Cro molecules produced in each 5-min imaging interval for all time traces (Fig. 3e). In addition to the two expected populations of high-CI ([L, H]), and high-Cro ([H, L]) production states, there were two additional populations. One resided at [0, 0] where no CI or Cro was produced, and another

centered at ~4 molecules for both CI and Cro, similar to the [L, L] and [H, H] populations we observed in steady-state measurements (Fig. 2). The one-dimensional (1D) histograms of CI and Cro alone showed two-state distributions (Fig. 3e).

We verified that the presence of the four populations was not caused by the independent production of CI and Cro from two copies of the λ switch due to chromosome replication, because the four populations existed similarly in young cells where the chromosomal copy was one before replication (cell age ⩽0.4, less than 40% of the cell cycle time, Supplementary Figure 10A). Furthermore, single-molecule fluorescence in situ hybridization (smFISH, Supplementary Table 3) showed co-existence of *cI* and *cro* mRNA molecules in a significant population of cells (16.3 ± 0.7%, *cI* and *cro* mRNAs at 0.9 ± 0.03 and 0.6 ± 0.02 molecules per cell, $\mu \pm$ s.e., $n = 2627$ cells), irrespective of cell ages (Supplementary Figure 10B and Supplementary Table 4). This result suggested that *cI* and *cro* mRNAs were produced within each other's short lifetime window (~1.5 min[35]) (Supplementary Note 4). Co-existence of *cI* and *cro* mRNAs has also been previously observed in cells growing under a different growth condition[35]. Finally, we verified that the stochastic maturation process of the Venus fluorophore only affected the spread, but not the presence, of each population in the 2D histogram (Supplementary Note 5, Supplementary Figure 11). Taken together, these results suggested that in each 5-min time window, a cell could produce none, only one or the other, or both proteins.

**Quantifying potential landscape and switching kinetics**. Using the experimentally measured 2D distributions of CI and Cro production levels, we generated the corresponding potential landscape by calculating the negative logarithm of probabilities (Fig. 3f). There were clearly four basins, approximately around at [0, 0], [4, 0], [0, 4], and [4,4] for produced [Cro, CI] protein molecule numbers per 5 min (Fig. 3f). Interestingly, there was one central peak separating the four basins such that the barrier height between two opposite basins [L, L] and [H, H], or [H, L] and [L, H], was higher than that between two adjacent basins [L, L] and [L, H], or [L, L] and [H, L] (Fig. 3f). This type of landscape has not been previously observed for such a genetic circuitry, and suggested specific switching paths between the basins. For example, to switch from [H, L] to [L, H], the path going through the [L, L] or [H, H] basins would have higher probability than the path of switching directly between the two.

To quantitatively identify possible production states of CI and Cro corresponding to the observed basins in the potential landscape, and obtain the associated transition rates between these states, we used a modified Hidden Markov Model (HMM) (Supplementary Note 1), which is commonly used in temporal pattern recognition[36,37]. We found that a four-state HMM ([L, L], [L, H], [H, L], and [H, H]) matched the observed 2D histogram of CI and Cro production the best (Supplementary Figure 12, Supplementary Note 2). The mean production levels of Cro and CI of each state and the corresponding dwell times were summarized in Table 1 and Supplementary Figure 13. Importantly, HMM allowed us to identify state-switching events in individual time traces (Fig. 3a–d, middle panels with colored bars) and hence the transition time constants (Supplementary Note 3, Fig. 4a, Supplementary Table 5)[17,38]. Similar results were observed using truncated time traces of only young cells (Supplementary Figure 14, Supplementary Tables 5 and 6).

Because the dynamics of a system is fully determined and described by its speed and the underlying kinetic processes (or paths), the transition time constants obtained here can be used to identify the most likely transition paths and the associated rates of switching between states, which has not been achieved before. For

example, to switch from the [L, H] state to the [H, L] state, the most likely path is to go through the [H, H] state instead of directly switching. We can also determine the time it takes for switching by the times a cell spent on the two paths (from [L, H] to [H, H] and from [H, H] to [H, L]). This gives us an insight into possible mechanisms underlying the kinetic processes in terms of the speed and the most likely paths, suggesting an unexpected

kinetic route through [H, H] beyond direct switching between CI and Cro.

## Discussion

Theoretical studies have shown that without changing the wiring configuration of a bistable switch, multistability can arise from

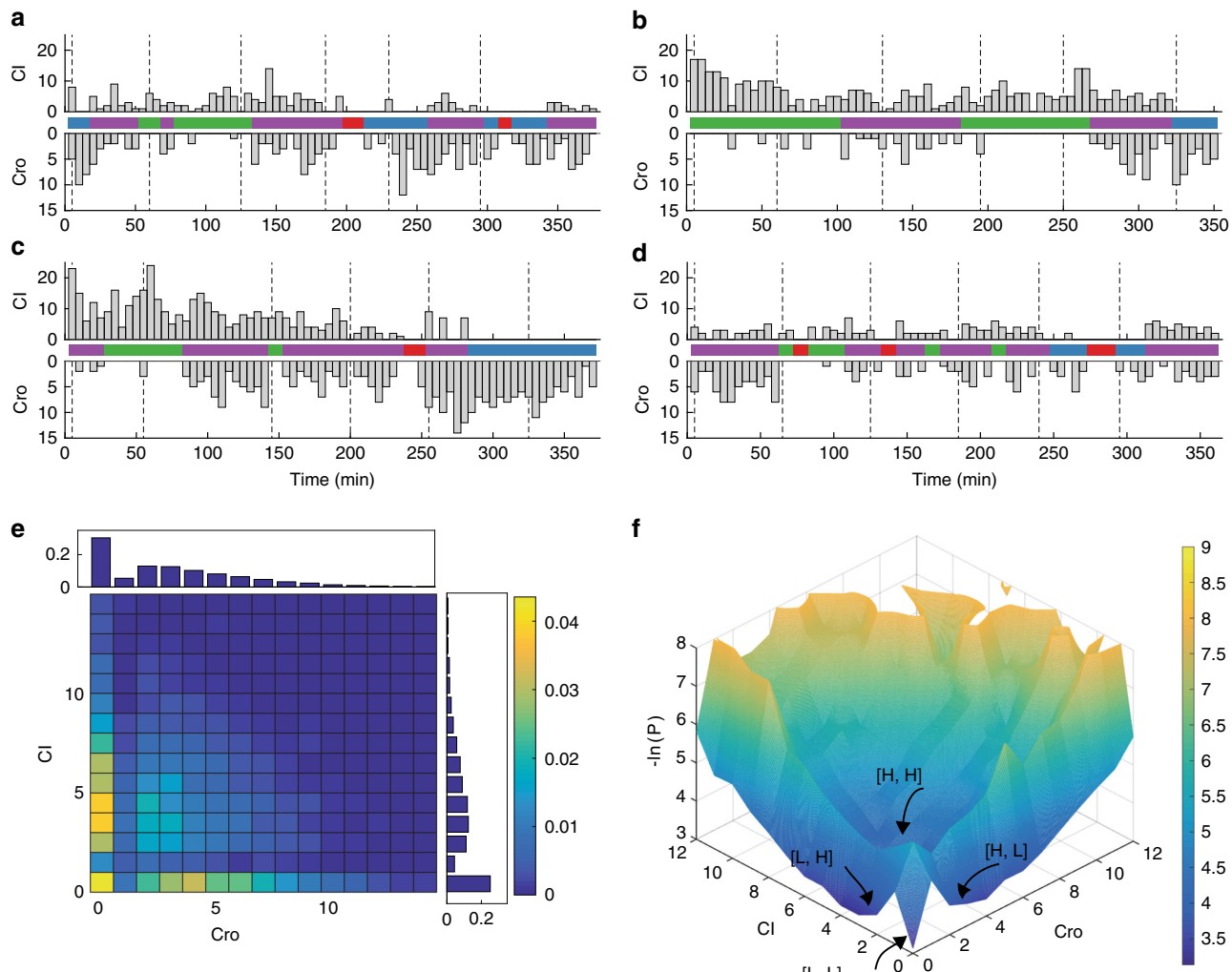

**Fig. 3** Real-time production time traces of CI and Cro in strain XF224 showed four distinct populations at 36.5 °C. **a–d** Time traces of newly produced CI (top) and Cro (bottom) molecule numbers in the same cells in four representative XF224 cell lineages. The corresponding state-switching time trace identified by HMM is shown in the middle panel of each lineage, with blue corresponding to state [H, L] for high Cro and low CI production, green to [L, H], purple for [H, H] and red for [L, L]. The dashed vertical lines indicate cell division. **e** 2D histogram of produced CI and Cro protein molecules in individual cells measured at each 5-min frame in time-lapse experiments ($n = 6453$ frames from 94 time traces). Corresponding 1D histograms of CI and Cro are shown on the right and top of the 2D histograms, respectively. Colors and scale bars indicate fractions of cells. **f** The potential landscape was calculated using the experimentally measured 2D histogram of CI and Cro expression numbers in every 5-min frame and interpolated

**Table 1 Mean production levels of CI and Cro and the corresponding dwell time of each state identified by HMM in time-lapse experiments**

| State [Cro, CI] | Cro[a] (molecules) | CI[b] (molecules) | $n$ (frames) | Dwell time[c] (min) | $n$[d] (occurrence) |
|---|---|---|---|---|---|
| [L, L] | 0.0 ± 0.01 | 0.0 ± 0.0 | 139 | 17 ± 2 | 41 |
| [H, L] | 5.2 ± 0.10 | 0.0 ± 0.0 | 1173 | 36 ± 3 | 162 |
| [L, H] | 0.0 ± 0.0 | 4.7 ± 0.13 | 1451 | 27 ± 1 | 269 |
| [H, H] | 4.5 ± 0.04 | 4.7 ± 0.05 | 3069 | 47 ± 3 | 396 |

[a, b, c]Values were expressed as mean ± standard error
[d]Number of occurrences of each state in all time traces

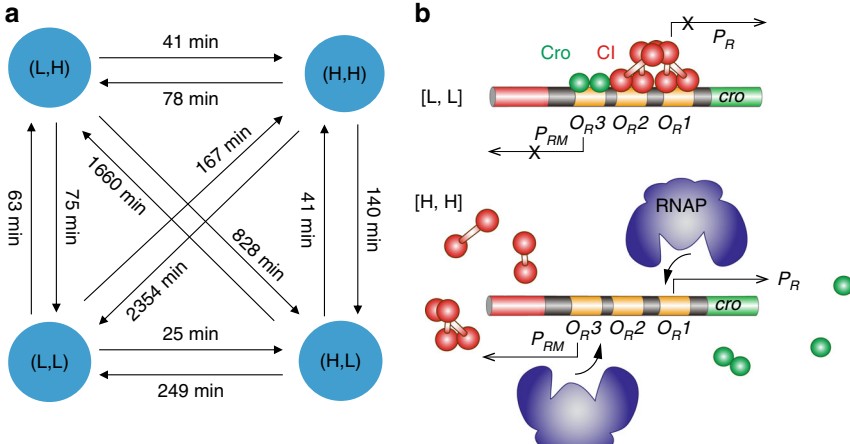

**Fig. 4** Transition time constants (**a**) and possible underlying molecular events (**b**) for the four stable production states observed in the λ switch. **a** Transition time constants were identified using a modified HMM analysis. Longer transition times indicate that direct switching between diagonal states is much less likely than that between side states. **b** The [L, L] states could arise due to the co-occupancy of the three operators by a combination of Cro and CI. The [H, H] state could arise from the slow association of Cro or CI to the operators, allowing RNAP to bind freely on $P_R$ or $P_{RM}$ to initiation transcription. Note that only these two simple possibilities were depicted here but other promoter configurations could also potentially lead to the [L, L] and [H, H] production states

weakened regulatory interactions, which impose fewer constraints on possible TF binding configurations[13–16]. In eukaryotic cells, epigenetic phenomena such as histone modification and DNA methylation could reduce the binding rates of TFs to their targeting DNA sites, leading to longer time scale of gene regulation. In bacterial cells, low TF expression levels[39] and high levels of non-specific binding[40] can effectively slow down the binding of TFs to specific target sites, leading to weakened regulation. Such a regime in gene regulation is termed non-adiabatic, in contrast to the classic adiabatic regime, in which protein binding and unbinding are fast compared to the protein's production and degradation time scales with rapid equilibration in a well-mixed environment[14–16,41]. Previous studies under different experimental conditions in which CI was expressed at higher levels than ours have elegantly demonstrated the adiabatic regime of the λ switch[8,42,43].

In strain XF224, both CI and Cro were expressed at much lower levels than in the WT strain XF204 (Supplementary Table 2, Supplementary Figures 5 and 15). Under this condition, slow association/dissociation times (in the range of a few minutes to hours[44,45]) and significant levels of non-specific binding for both CI and Cro (>70%,[40]) were previously demonstrated. Our results showed that the [L, L] state persisted for ~20 min (Fig. 3a–d), which is longer than the reactions times typically associated with relevant biochemical events such as transcription initiation, mRNA degradation, and TF binding[46]. The [L, L] production state could emerge and remain relatively long-lived when a combination of CI and Cro occupies all the operator sites, shutting down the production of CI and Cro simultaneously (Fig. 4b).

When either CI or Cro dissociates, and the rebinding is slow, RNA polymerase (RNAP) can bind to the exposed $P_R$ or $P_{RM}$ promoters, resulting in the [H, L] or [L, H] production states. RNAP exists at a much higher level (~3000 molecules per cell under a similar growth condition[47]) compared to CI and Cro, and hence its binding rate would be significantly faster. When both CI and Cro dissociate from all three operators, which would occur at a much lower probability than only one of them dissociating, RNAP can initiate transcription from either one of the two promoters, resulting in the [H, H] production state (Fig. 4b). Consistent with this possibility, we previously measured the basal

expression level of $P_{RM}$ promoter in the absence of CI and Cro to be similar to the CI production level in the [H, H] state[27]. In addition, in vitro studies have demonstrated that both $P_R$ and $P_{RM}$ promoters on the same λ switch can be occupied by two RNAP molecules simultaneously in the absence of CI and Cro[48,49].

One important aspect of our time-lapse experiments, in contrast to an earlier experiment with a similar genetic network[8], is that we measured the production, but not cellular concentrations, of CI and Cro. Protein production rates directly measure promoter activities, which reflect promoter configurations with respect to TF and RNAP binding. In the adiabatic regime, protein concentration changes (and hence changes in protein binding rates) are instantaneously reflected in protein production rates; in the non-adiabatic regime, however, promoter activity can lag protein concentration changes. One concentration state can correspond to multiple production states, and hence multiple cell fate potentials. The well-known hysteresis effect of bistable switches[8] is likely a result of the non-adiabatic cellular environment in which protein binding/unbinding is slow—cells starting in one state have a tendency to stay in that state before switching to the other states even when the concentration of a critical protein has already changed.

Previous studies showed that different wiring conditions of bistable switches could give rise to a maximum of three states in the adiabatic regime[17–19]. Here we showed that, in the non-adiabatic regime, four protein production states can emerge from bistable switches without changing wiring configurations, with consequences in establishing new cell fates[13–16,50]. A living cell, in which a non-equilibrium state is the norm, could potentially utilize non-adiabaticity to encode more than two cell fates with limited circuitry, allowing a high degree of adaptation and differentiation. However, from the opposing point of view, this increased diversity of states in the non-adiabatic regime places more limits on genetic circuitry that will produce robust binary switching; hence, avoiding the non-adiabatic regime will be key to engineer robust, binary genetic switches.

## Methods

**Bacterial strains and plasmids construction**. Strains XF103 and ZH051 were generated using the parental strain BW25113[51] and λ RED recombination[52].

Briefly, plasmid pXF103 carrying the *lacI-venus-ub-cI* fusion gene was constructed by replacing the *tsr-venus* sequence of the λ switch in plasmid pZH051[27] with the *lacI-venus* sequence amplified from plasmid pVS143 using primers P1 and P2. The full-length λ switch sequence (from $O_L$ to the end of *cro*) containing *lacI-venus-ub-cI* on pXF103, or *tsr-Venus-ub-cI* on pZH051, was then PCR amplified (primer pair P3:P4) together with the drug resistance gene *kan^R* on the plasmid and replaced the *lacI* gene on chromosome using λ RED recombination. Subsequently, the *kan^R* gene was removed by transforming the FLP recombinase expressing plasmid pCP20 into the host strain to generate the final XF103 or ZH051 strain.

Strains XF204, XF016, XF206, XF214, XF224, XF225, and XF226 were generated using the parental strain *E. coli* K12 MG1655 (Yale University *E. coli* Genetic Stock Center) and the landing pad approach, which is specific for the chromosomal integration of large genetic constructs[53]. First, the *ind1* and *sam7* mutations in the *cI857* sequence on plasmid pZH016 were corrected using primers P5 and P6, and the resulting *tsr-venus-ub-cI857* was used as a template for subsequent strain constructions. To generate the two-reporter λ switch (*tsr-venus-ub-cI* in place of *cI* and *lacI-venus-ub-cro* in place of *cro*), *lacI-venus-ub* was amplified from plasmid pXF103 using primer pair P7:P8 and ligated into pZH016 in front of *cro* to generate two-reporter λ switch pXF104. The landing pad vector pTKIP (containing LP1 and LP2, gift from Dr. Thomas E. Kuhlman) was opened to add NheI and SalI restriction sites at two ends using inverse PCR (primer pair P9: P10). Plasmid pXF104 was digested with NheI and SalI to release the two-reporter λ switch DNA fragment and ligated into similarly digested pTKIP inverse PCR product to obtain plasmid pXF204. Plasmid pXF204 was then served as a template to generate pXF214 (primer pair P11:P12, ATG start codon of *lacI* changed to GTG) and pXF224 (primer pair P13:P14, $P_R$ promoter −32 A to G) using site-directed mutagenesis.

To eliminate the fluorescence of Venus, two glycine residues in the tri-peptide of Venus chromophore were mutated to Alanin[54] using site-directed mutagenesis (primer pair P15:P16) to obtain pXF106 from pZH016. Plasmid pXF106 was digested by BspEI and SalI to release the *tsr-venus*(G65A and G67A)-ub-cI857* fragment and ligated with similarly digested vector from pXF204 to obtain pXF206. Plasmid pXF106 was digested by BspEI and SalI to release the *tsr-venus*(G65A and G67A)-ub-cI857* fragment and ligated with similarly digested pXF224 to obtain pXF226, which contain the *tsr-venus*-ub-cI* and the mutated $P_R$ promoter in front of *lacI-venus-ub-cI*.

To generate pXF016, which was the landing pads version of pZH016, pZH016 was digested with NheI and SalI to release the λ switch DNA fragment containing *tsr-venus-ub-cI857* and *cro*. The fragment was then ligated into NheI- and SalI-digested pXF214 to obtain pXF016. pXF225 was generated from pXF016 using site-directed mutagenesis (primer pair P13:P14) to mutate $P_R$ promoter (−32 A to G).

To prepare the parental strain MG1655 for landing pad integration, the fragment of LP1-*tet^R*-LP2 containing two landing pads LP1, LP2, and a tetracycline resistance gene was amplified from plasmid pTKS/CS (gift from Dr. Thomas E. Kuhlman). The *lacI* gene on MG1655 chromosome was then replaced with the LP1-*tet^R*-LP2 fragment using λ RED recombination to obtain strain XF001. To construct strain XF204 using the landing pad approach, plasmids pTKRED (gift from Dr. Thomas E. Kuhlman) and pXF204 were transformed into XF001. Single colonies of XF001(pTKRED/pXF204) transformants were picked and grown in 1 ml LB with 2% arabinose and 4 mM IPTG at 30 °C for 2 h with aeration. Next, 10 μl spectinomycin at a final concentration of 100 μg ml$^{-1}$ was added and the culture was allowed to continue at 30 °C. After 5 h, 1 μl kanamycin at a final concentration of 50 μg ml$^{-1}$ was added and the culture was kept growing overnight. The next morning, the overnight culture was 1:10$^4$ diluted using fresh M9 medium, and 50 μl of the dilution was plated on LB plate with kanamycin (50 μg ml$^{-1}$) and incubated overnight at 30 °C. Single colonies grown on the kanamycin plate were tested for their failure to grow in tetracycline- or carbenicillin-containing media and subsequently sequenced to obtain strain XF204. Correct colonies were picked into 5 ml LB medium and cultured overnight at 37 °C to eliminate plasmid pTKRED. The other strains (XF016, XF206, XF214, XF224, XF225, and XF226) were constructed following the same procedure using the corresponding helper plasmid (pXF016, pXF206, pXF214, pXF224, pXF225, and pXF226). All the strains were then transformed with the UBP1-expressing plasmid pCG001 (gift from Dr. Roland Baker[55]) and the *lacO^256* plasmid pZZ6 (gift from Dr. Joe Pogliano[30]). Note that the 37 °C growth condition to eliminate pTKRED led to a large population of cells expressing high levels of Cro in the presence of the CI857 mutant, especially in the XF204 strain. Therefore, after the elimination of pTKRED, plasmid cells were grown in LB medium at room temperature overnight followed by restaking on LB plates at 30 °C for another day's growth. Single colonies that have already switched back to low Cro expression levels (lower fluorescence level compared to that in high Cro expression levels) were identified using a Halogen lamp equipped with an emission filter (ET545/30, Chroma). These colonies were further grown to exponential phase in M9 medium at 30 °C and imaged on the microscope to confirm their expression states. Confirmed cultures were then flash-frozen in liquid nitrogen and stored at −80 °C.

To construct plasmid pXF011 that expressed UBP1 and LacI-mCherry, *lacI-mCherry* fragment with the pBAD promoter was PCR-amplified from pZH102 using primer pair P19:P20, subsequently restricted with SalI and EagI, and ligated to similarly restricted pCG001 vector to obtain plasmid pXF004. The pBAD promoter in front of *lacI-mCherry* on pXF004 was then replaced by a constitutive promoter of BBa_J23103[56] using primer pair P21:P22 to obtain pXF011.

All the strains, plasmids, and primers are listed in Supplementary Table 1.

**Growth conditions**. Cells from frozen stocks were streaked on LB plates and incubated at an appropriate temperature overnight. Single colonies were picked the next day and inoculated into M9 medium supplemented with MEM amino acid (Sigma-Aldrich Co. LLC) at appropriate temperature overnight in a precision temperature-controlled shaker (±0.5 °C, MIDSCI IS-300). The next morning, cells were reinoculated in fresh M9 medium to mid-log phase (OD$_{600}$≈0.4) before steady-state or time-lapse microscopy experiments. Antibiotics were included in all cultures at concentrations of 50 μg ml$^{-1}$ for kanamycin, 25 μg ml$^{-1}$ for chloramphenicol, and 100 μg ml$^{-1}$ for ampicillin when appropriate.

**Western blot**. Cells were cultured under the same condition as that described for steady-state microscopy measurements. Log phase cells (OD$_{600}$≈0.4–0.5) were collected and cell numbers counted using a Petroff-Hausser chamber and a plating assay. Cell lysates were prepared by incubating equal number of cells for 10 min at 100 °C followed by 20 min at −75 °C. Protein electrophoresis was carried out using a 4–15% Tris-HCl Precast gradient gel (Bio-Rad) at 100 V for 1.5 h. The gel was then transferred to a PVDF membrane (Bio-Rad) for 2 h at 25 V and 4 °C. Venus bands were detected with 1:2500 mouse antibody to GFP (Clontech, JL-8) and 1:20,000 goat-anti-mouse HRP secondary antibody (BioRad, #170-5047). ImmunStarTM WesternC^TM reagents (Bio-Rad) were applied for luminescent visualization. Images were captured using a Typhoon Scanner (GE Life Sciences).

**smFISH**. CI transcripts were labeled with 30 oligonucleotides (Supplementary Table 3) conjugated with TAMRA (Biosearch Technologies). Cro transcripts were labeled with 41 oligonucleotides (Supplementary Table 2) conjugated with Quasar 670 (Biosearch Technologies). Because of the short sequence of *cro* mRNA, only nine of the probes targeted to the *cro* mRNA sequence and 31 probes targeted to the *lacI* sequence fused to *cro*.

Three cultures of XF224 cells were grown using the same procedure and growth medium as that for the time-lapse experiments, but at three different temperatures, 30, 36.5, and 37.5 °C. The 30 °C culture was used as a control to quantify the fluorescence intensity of single *cro* mRNA molecules, because at 30 °C CI dominated and Cro was transcribed at such a low level that most *cro* transcripts were single molecules. Similarly, the 37.5 °C culture was used as a control to quantify the fluorescence intensity of single *cI* mRNA molecules. Cells were fixed at respective growth temperatures and labeled with 1 μM CI and 1 μM Cro probes using a protocol as described previously[57]. Briefly, the cells were fixed for 30 min with 3.7% formaldehyde and were then permeabilized with 70% ethanol for 1 h. Each sample was hybridized overnight in a 40% formamide hybridization solution. Before imaging, the cells were washed 4× with 40% formamide wash solution and then resuspended in 2x SSC for imaging.

Fixed and labeled cells were imaged using simultaneous laser excitation at 561 nm (Coherent, sapphire) and 647 nm (Coherent, obis). Emission was collected using an OptoSplit III with a long-pass (647 nm) beam splitter and emission filters ET590/33 and HQ705/55 (Chroma Technology). Each view field was imaged at six z planes separated by 200 nm. The projection of the six planes was then used to detect fluorescent spots using custom MATLAB software as previously described[30]. The total integrated fluorescence intensity of each spot was divided by the mean intensity of corresponding single transcript molecules to obtain the number of transcript molecules in each spot.

**Time-lapse imaging**. Log phase cell culture (1 ml) was collected and washed twice with fresh M9 medium. Pelleted cells were diluted 1:100 and 0.5 μl was spotted onto a gel pad made of 3% low-melting temperature SeaPlaque^TM agarose (Lonze) using M9 medium in a precision temperature-controlled growth chamber (FCS2, Bioptechs). The chamber was locked on an Olympus IX-81 inverted microscope equipped with a 100× oil-immersion objective lens (Olympus Inc., PlanApo 100×NA 1.45). Both the sample chamber and the objective were maintained at 36.5 °C using respective heaters provided by the FCS2 system (Bioptechs). In all time-lapse experiments, a digital thermometer probe was inserted into the sample chamber to record the temperature fluctuations in real time using a voltage recorder (MadgeTech, VOLT101A-15V). Excitation at 514 nm with an illumination power density of ~1 kW cm$^{-2}$ was provided by an argon ion laser (Coherent Innova I-308). Emission was filtered (ET545/30, Chromas Technology Corp), and fluorescent and bright-field images with a time interval of every 5-min were captured by an Ixon EMCCD camera (Andor IXon DU888) using a custom imaging journal in Metamorph (Molecular Devices) as previously described in refs. [26,27].

**Time-lapse image analysis**. We used a previously published procedure to segment cells, detect fluorescent spots, and track cell lineages[27]. We used a custom Matlab code to assign individual fluorescent spots to CI or Cro by measuring the centroid distance of the spot to cell center (estimated as the center of mass of cells in a segmented, binary image) using a threshold of 0.6 (Fig. 1c and Supplementary Figure 7). For each micro-colony, only one cell lineage time trace with complete cell cycles was used. A total of 96 time traces (6453 frames) were obtained for the XF224 strain.

**Steady-state imaging**. Log phase cell culture was prepared exactly the same as described in time-lapse imaging except that cells were diluted 1:50 to obtain more cells in each imaging area. All cells were imaged within 1.5 h at room temperature to avoid significant changes of expression levels. For the two-color strain XF004, Venus was excited using an argon ion laser (Coherent Innova I-308) and mCherry was excited by a rhodamine dye laser (Coherent 599) tuned to ~570 nm. Images were split into the yellow and red channels by an Optosplit II adaptor (Andor) using a long-pass filter, and further filtered by ET545/30 and HQ630/60 bandpass filters (Chroma) for the yellow and red channels, respectively.

**Steady-state image analysis**. To quantify the copy numbers of CI and Cro in microscopy experiments, we measured the fluorescence intensity distribution of single Tsr-Venus-Ub molecules under our imaging condition (Supplementary Figure 4) using the $\lambda^-$ strain expressing very low numbers of Tsr-Venus-Ub per cell cycle[27]. Using a previously described procedure, individual, well-localized Tsr-Venus-Ub or LacI-Venus-Ub spots in experimental strains were detected and the corresponding fluorescence intensity was converted to molecule numbers of CI or Cro by dividing by the peak value of the Gaussian distribution of single Venus molecules[27]. For strain XF224, both Tsr-Venus-Ub and LacI-Venus-Ub molecules were localized to diffraction-limited spots and the procedure described above was carried out. For strains XF204 and XF214, at high temperatures, Cro was expressed at high levels, and hence did not form well-localized spots but nucleoid-covered clouds. Therefore, we measured total integrated cellular fluorescence and subtracted the fluorescence of pole-localizing Tsr-Venus-Ub spots. The subtracted fluorescence intensity was then used to calculate the number of expressed Cro copy numbers. The absolute copy numbers of the three strains at different temperatures measured at steady state are plotted in Supplementary Figure 5 and summarized in Supplementary Table 2.

**Code availability**. The analyses in this study were performed by using custom MATLAB code, which will be made available from the corresponding author upon reasonable request.

**Data availability**. The authors declare that the data supporting the findings of this study are available from the corresponding author upon reasonable request.

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

## Acknowledgements

We thank Dr. Thomas E. Kuhlman for the gifts of pTKRED, pTKIP, pTKS/CS; Dr. Roland Baker for the gift of pCG001; Dr. Joe Pogliano for the gift of pZZ6. This work was supported by NSF CAREER Award (0746796), March of Dimes Research grant (1-FY2011), NSF EAGER MCB1019000, NSF PHYS 76066, NSFC 91430217, and LISBOA-01-0145-FEDER-007660.

## Author contributions

X.F., J.W. and J.X. designed the projects. X.F. engineered the strains, performed immunoblotting experiments, acquired and analyzed the fluorescence images. Q.L., Z.H. and J.X. developed the image data processing method. Q.L., W.H. and J.W. developed the analytical method. C.B. performed smFISH experiment. Q.L., J.W. X.F. and J.X. analyzed the data. X.F., J.X., and J.W. wrote the manuscript.

## Additional information

**Competing interests:** The authors declare no competing interests.

