## [Peer Review File · Nature Communications]

Reviewers' comments:

Reviewer #1 (Remarks to the Author):

In the article, the authors creatively developed a dual-gene reporting system that uses co-translational cleavage and different subcellular localizations of the same reporting protein to facilitate the simultaneous observation of the expression of two genes. Based on this method, they examined the Cro-CI bistable switch system, and observed non-canonical state(s) previously not reported. They further followed the production of the two transcriptional factors, and developed a potential landscape of the different states. With the new observations, the authors argue that in the non-adiabatic regime, novel stable states could emerge from the canonical bistable switches.

Overall, this is an important and timely study that uncovers new regimes and dynamic behaviors of a classical genetic toggle switch. In the study, experiments were elegantly designed and well implemented, theoretical analyses were seamlessly integrated, and the results and conclusions were well presented. I think it will have broad impacts in the fields of systems biology and synthetic biology, and will interest readers with diverse background.

I would warmly recommend publication with the following relative minor comments being addressed:

1. Although this is a fundamental study that focuses on the discovery and analysis of the new states of the circuits, it provides valuable insights for the field of synthetic biology which focuses on the development of novel cellular functions with engineered gene circuits. From an engineering perspective, an ideal switch circuit will have only the two mutually exclusive states, (H,L) and (L,H), for robust and reliable cellular state switching. The other two states are naturally existing but may doom the key switching function of the circuit. I thus encourage the authors to include a discussion about the implications of this study for the design and engineering of synthetic gene circuits: to enable robust practical biotechnological applications by minimizing unwanted dynamic regimes.

2. In line 203, the authors referred to Extended Data Fig.15 for expression levels in XF224. It would be helpful to have the WT levels for comparison.

3. In line 206, the authors claimed that the [L, L] production state can be stable. However, the reasoning doesn't appear very convincing. The authors argued that this could result from the condition when both promoters are occupied by the TFs. But it would be reasonable to imagine that as time goes on, TFs would be degraded or diluted, their concentrations would drop (as there's little new production), and the probability of both promoters continuing to be bound would decrease as well, eventually the cell would enter another state different from [L, L]. To make a stronger case, the authors can consider developing a more quantitative claim, or modifying the claims otherwise.

4. In line 105, the authors claimed that the cells were maintained at steady-state. The authors need to clarify the experimental meaning of this claim. In the Methods section, it doesn't appear that the authors tried to maintain fresh medium for the cell growth during observation, and the length of observation was mentioned to be 1.5 hrs, which doesn't appear long enough for us to assume that the statistical distributions of the cell states won't change as time goes on and on (even if we count the pre-observation inoculation time).

Reviewer #2 (Remarks to the Author):

Fang et al. studied a classic bistable-switch system, phage lambda's CI-Cro at the single-molecule

level. They developed a novel single-molecule live-cell imaging technique to examine CI and Cro protein levels simultaneously. Based on the experimental measurements, they built a mathematical model and calculated the switching parameters between different states. The major finding is this bistable-switch system has 4 states ([H L], [L H] and two new states [H H] and [L L]) instead of the traditional 2-state ([H L] and [L H]) in a non-adiabatic gene-regulation regime, and the switching from [H L] state to [L H] state is more likely through a middle step [H H] instead of directly switching between. I found this paper interesting and well-written. I have a few comments below.

1) My concern is that the key experiments were done at 36.5 C using cI857, where they observed the two new gene expression states, or the 'new' cell fate potentials. At 36.5 C, the activity of CI is hard to predict, maybe each CI protein is at a sub-optimal state and cannot bind well, or maybe a subpopulation of CI is fully inactivated while the others are fully functional. The 'new' gene expression states seem to be a result of this weird CI state (a lot of uncertainties going on here). I'd like the authors to address/discuss this question.

2) This paper only addresses the immunity region of OR, what about the contribution from OL?

3) In line 203-219, the authors were trying to explain the rise of the [H, H] state was due to the lower unbinding rate of CI/Cro comparing with production rate. The literature (43 and 44) showed the disassociation and the rebinding of CI and Cro to the promoters are slow. But a recent paper (Sepulveda et al., Science 2016) suggests switching between promoter configurations is fast (the CI binding and unbinding should be fast). I'd like the authors to address this question.

4) The 4 different states were observed at different temperatures (30, 36.5 and 37C). We know that the physiology of the cell is highly dependent on temperature. How does the cell physiology play a role here?

5) When analyzing the cell lineages, have all lineages from the same mother cell included? Any difference between daughter cells? Any correlation between different cell generations and the distribution (Figure 3)?

Minor comments.

1). line 204. In the range of a 'few' minutes to ...

2). Figure 1, panel B. the lower right cell seems to have its focus at the pole. For representative images, it should be good to avoid these confusions.

3). Figure 2, it will help the readers if the legends were included in panel A, and scale bars in panel B.

4). Include time stamp and scale bar in the videos.

Response to Reviewers' comments:

Reviewer #1 (Remarks to the Author):

In the article, the authors creatively developed a dual-gene reporting system that uses co-translational cleavage and different subcellular localizations of the same reporting protein to facilitate the simultaneous observation of the expression of two genes. Based on this method, they examined the Cro-CI bistable switch system, and observed non-canonical state(s) previously not reported. They further followed the production of the two transcriptional factors, and developed a potential landscape of the different states. With the new observations, the authors argue that in the non-adiabatic regime, novel stable states could emerge from the canonical bistable switches.

Overall, this is an important and timely study that uncovers new regimes and dynamic behaviors of a classical genetic toggle switch. In the study, experiments were elegantly designed and well implemented, theoretical analyses were seamlessly integrated, and the results and conclusions were well presented. I think it will have broad impacts in the fields of systems biology and synthetic biology, and will interest readers with diverse background.

I would warmly recommend publication with the following relative minor comments being addressed:

1. Although this is a fundamental study that focuses on the discovery and analysis of the new states of the circuits, it provides valuable insights for the field of synthetic biology which focuses on the development of novel cellular functions with engineered gene circuits. From an engineering perspective, an ideal switch circuit will have only the two mutually exclusive states, (H,L) and (L,H), for robust and reliable cellular state switching. The other two states are naturally existing but may doom the key switching function of the circuit. I thus encourage the authors to include a discussion about the implications of this study for the design and engineering of synthetic gene circuits: to enable robust practical biotechnological applications by minimizing unwanted dynamic regimes.

Thanks for the great suggestion. Indeed for synthetic biology the extra states in the non-adiabatic regime may bring in unwanted consequences. We now added new discussion in the last paragraph of the article raising this awareness.

2. In line 203, the authors referred to Extended Data Fig.15 for expression levels in XF224. It would be helpful to have the WT levels for comparison.

We measured the expression levels for the WT strain XF204 and added the comparison with that of XF224 in line 202-203. The results were also shown in Extended Data Table 2, Extended Data Fig. 15 and Extended Data Fig. 15.

3. In line 206, the authors claimed that the [L, L] production state can be stable. However, the reasoning doesn't appear very convincing. The authors argued that this could result from the condition when both promoters are occupied by the TFs. But it would be reasonable to imagine that as time goes on, TFs would be degraded or diluted, their concentrations would drop (as there's little

new production), and the probability of both promoters continuing to be bound would decrease as well, eventually the cell would enter another state different from [L, L]. To make a stronger case, the authors can consider developing a more quantitative claim, or modifying the claims otherwise.

We agree that the [L, L] state will eventually switch into other states, as any of the other three states will do. For example, the [L, H] state for low Cro and high CI production can switch to the [L, L] state when CI or a combination of CI and Cro occupies all the three operators. A cell at a particular state has a certain lifetime and transition rate (Figure 4, Table 1, Extended Table S5) to move out of the current state and into another state. In this sense, these states are not permanent or mega-stable, but are local minima that a cell can stay in for a relatively long time. Accordingly, we changed the term from “stable” state to “relatively long-lived” state and added new discussions in the main text (see lines 202 to 211).

4. In line 105, the authors claimed that the cells were maintained at steady-state. The authors need to clarify the experimental meaning of this claim. In the Methods section, it doesn't appear that the authors tried to maintain fresh medium for the cell growth during observation, and the length of observation was mentioned to be 1.5 hrs, which doesn't appear long enough for us to assume that the statistical distributions of the cell states won't change as time goes on and on (even if we count the pre-observation inoculation time).

The length of observation in time-lapse experiments on microscope was > 5 generations (or > 350 min) for most cell lineages. In general, our growth condition was to inoculate cells in M9 medium overnight in a precision temperature chamber with aeration to an O.D. of 0.6 – 0.8, and the next morning to dilute the cultures down with fresh M9 medium to allow further growth. We then collect cells at O.D. of ~ 0.4 for time-lapse or snapshot measurements. Therefore, cells were maintained at one constant temperature for > 20 generations. We have previously shown that using this growth condition and the gel pad growth chamber, cells can grow steadily on microscope stage with constant expression levels, sufficient nutrients and oxygen for > 350 mins^{1,2}. Therefore, we consider that these cells have reached steady-state. We recognize that this growth condition is different from the classic “steady-state” growth by growing cells in a chemostat. We have modified our text to clarify this point so to avoid confusion with the classical steady-state definition in the main text (lines 91 to 94, and 108 to 115).

Reviewer #2 (Remarks to the Author):

Fang et al. studied a classic bistable-switch system, phage lambda's CI-Cro at the single-molecule level. They developed a novel single-molecule live-cell imaging technique to examine CI and Cro protein levels simultaneously. Based on the experimental measurements, they built a mathematical model and calculated the switching parameters between different states. The major finding is this bistable-switch system has 4 states ([H L], [L H] and two new states [H H] and [L L]) instead of the traditional 2-state ([H L] and [L H]) in a non-adiabatic gene-regulation regime, and the switching from [H L] state to [L H] state is more likely through a middle step [H H] instead of directly switching between. I found this paper interesting and well-written. I have a few comments below.

1) My concern is that the key experiments were done at 36.5 C using cI857, where they observed the two new gene expression states, or the 'new' cell fate potentials. At 36.5 C, the activity of CI is hard to predict, maybe each CI protein is at a sub-optimal state and cannot bind well, or maybe a subpopulation of CI is fully inactivated while the others are fully functional. The 'new' gene expression states seem to be a result of this weird CI state (a lot of uncertainties going on here). I'd like the authors to address/discuss this question.

Partially active CI indeed will be a concern. We carefully selected the CI⁸⁵⁷ mutant because it has been well characterized previously. Biochemical Investigations have shown that the destabilized CI⁸⁵⁷ loses its full functionality because it became mis-folded and degraded, and that temperature only changes the fraction of active/inactive CI molecules, but not the level of activity of individual CI molecules^{3,4}. We modified the text to further clarify this point (Lines 82-83).

2) This paper only addresses the immunity region of OR, what about the contribution from OL?

In all constructs used in this study we integrated the full-length λ switch sequence (from O_L to the end of *cro*) into the *E. coli* chromosome (Line 75). For simplicity the O_L was not shown in the figures, as it was the same for all the constructs. At the low production levels (~ 2-3 CI molecules per 5 min and fast degradation time), we expect that the looping frequency between O_L and O_R would be minimal, as we previously measured in⁵, thus contributing minimally to the observed states.

3) In line 203-219, the authors were trying to explain the rise of the [H, H] state was due to the lower unbinding rate of CI/Cro comparing with production rate. The literature (43 and 44) showed the disassociation and the rebinding of CI and Cro to the promoters are slow. But a recent paper (Sepulveda et al., Science 2016) suggests switching between promoter configurations is fast (the CI binding and unbinding should be fast). I'd like the authors to address this question.

In our original manuscript we discussed the work by Sepulveda et al., We believe that work elegantly demonstrated the switching kinetics under the adiabatic regime. There are two major differences between the two studies. First, the work by Sepulveda et al., used a different growth condition (rich complex medium LB, 37 °C v.s. our M9 minimal medium and 36.5 °C), and observed a significant higher expression number of CI protein (~ 190 molecules present per cell v.s. our ~ 50 molecules produced per cell) and mRNA (~ 6 mRNA/cell v.s. our ~ 1 mRNA/cell). The high expression level of CI favors the adiabatic regime. Second, In Sepulveda et al., there is very little Cro expression, which does not allow the observation of switching as we did in our work. We believe that Sepulveda et al., monitored one single state in which CI is expressed highly and Cro expressed lowly, corresponding to our [L, H] state. We have added discussion in lines 198 to 210 and 226 to 234 to further clarify these points.

4) The 4 different states were observed at different temperatures (30, 36.5 and 37C). We know that the physiology of the cell is highly dependent on temperature. How does the cell physiology play a role here?

For strain XF224, we observed the presence of all four states at the same temperature 36.5 °C. However, we were indeed concerned about possible physiological difference of cells at different temperatures, and constructed two additional strains XF204 and 214. These two strains had different expression levels of Cro and both showed the presence of the four states at one constant temperature (33 °C and 34.5 °C respectively, Figure. S5). These experiments demonstrated that the presence of the four states is not dependent on a particular temperature, but a general phenomenon.

5) *When analyzing the cell lineages, have all lineages from the same mother cell included? Any difference between daughter cells? Any correlation between different cell generations and the distribution (Figure 3)?*

These questions are very intriguing. For our data analysis in this work, we only picked one lineage from one colony to avoid data duplication in the correlation and HMM analyses. As such all the lineages used in the work had different mother cells. We have analyzed cell-cycle dependence of the distribution of CI and Cro production and did not see a significant difference. In the future we plan to analyze the correlation/difference between daughter cells, which will be the subject of another study. We added a sentence to clarify how we did the analysis (Line 131)

Minor comments.

1). *line 204. In the range of a 'few' minutes to ...*

Corrected.

2). *Figure 1, panel B. the lower right cell seems to have its focus at the pole. For representative images, it should be good to avoid these confusions.*

Thanks for the suggestion. We selected another representative image.

3). *Figure 2, it will help the readers if the legends were included in panel A, and scale bars in panel B.*

Thanks for the suggestion. We added the legends and scale bars.

4). *Include time stamp and scale bar in the videos.*

Thanks for the suggestion. We added the time stamp and scale bars.

References

- 1 Hensel, Z. *et al.* Stochastic expression dynamics of a transcription factor revealed by single-molecule noise analysis. *Nat Struct Mol Biol* **19**, 797-802, doi:10.1038/nsmb.2336 (2012).
- 2 Hensel, Z. & Marquez-Lago, T. T. Cell-cycle-synchronized, oscillatory expression of a negatively autoregulated gene in E. coli. *eprint arXiv*, 1506.08596 (2015).

- 3 Hecht, M. H., Nelson, H. C. & Sauer, R. T. Mutations in lambda repressor's amino-terminal domain: implications for protein stability and DNA binding. *Proc Natl Acad Sci U S A* **80**, 2676-2680 (1983).
- 4 Bednarz, M., Halliday, J. A., Herman, C. & Golding, I. Revisiting bistability in the lysis/lysogeny circuit of bacteriophage lambda. *PLoS One* **9**, e100876, doi:10.1371/journal.pone.0100876 (2014).
- 5 Hensel, Z., Weng, X., Lagda, A. C. & **Xiao, J.** Transcription-factor-mediated DNA looping probed by high-resolution, single-molecule imaging in live E. coli cells. *PLoS Biol* **11**, e1001591, doi:10.1371/journal.pbio.1001591 (2013).

REVIEWERS' COMMENTS:

Reviewer #1 (Remarks to the Author):

The authors have addressed all of my concerns in the revised manuscript. The quality of the work has been greatly improved. I warmly recommend it accepted for publication.

Reviewer #2 (Remarks to the Author):

One minor suggestion for the authors is: change the superscript 857 to subscript or normal font when spelling out cI857 allele to obey the convention in phage biology.

I am satisfied that the authors have addressed all my concerns. I believe this work is fit for publication and would be of interest to a broad scientific community.

Response to Reviewers' comments

Reviewer #2 (Remarks to the Author):

One minor suggestion for the authors is: change the superscript 857 to subscript or normal font when spelling out cI857 allele to obey the convention in phage biology.

We have changed all CI⁸⁵⁷ to CI₈₅₇.